# The Impact of Prenatal Environmental Tobacco Smoking (ETS) and Exposure on Chinese Children: A Systematic Review

**DOI:** 10.3390/children10081354

**Published:** 2023-08-07

**Authors:** Huazhen Ye, Xiaoyu Yang, Fahad Hanna

**Affiliations:** Public Health Program, Department of Health and Education, Torrens University Australia, Melbourne, VIC 3000, Australia; kfwxgw@gmail.com (H.Y.); xiaoyu.yang@health.torrens.edu.au (X.Y.)

**Keywords:** prenatal environmental smoking, child health, Chinese children, systematic review, adverse events

## Abstract

**Background:** There is considerable evidence to support the association between exposure to environmental tobacco smoke (ETS) and children’s burden of disease. However, the literature on the health outcomes of prenatal ETS exposure among Chinese children has not yet been comprehensively reviewed. Objective: This systematic review examines the currently available evidence and identifies gaps for further research on the health consequences of prenatal ETS exposure on Chinese children. **Methods:** Following the JBI systematic-scoping review methodological framework, we conducted a computer-aided search of three electronic databases—PubMed, EBSCOhost, and ProQuest to include studies from January 2011 to May 2023 that addressed the health outcomes of Chinese children whose mothers were exposed to ETS at any stage of pregnancy. Furthermore, a methodological quality assessment of the selected articles was conducted using JBI critical appraisal checklists. **Results:** A total of 30 articles were reviewed, including eleven high-quality studies and nineteen moderate-quality studies. Five main themes, including hypertension, fetal and children’s development, behavioural disorders, respiratory outcomes, and “other health outcomes”, were encompassed. The majority of the studies showed a positive link between prenatal ETS exposure and an increased risk of preterm birth, and moderate risk of fetal growth restriction. A few studies explored other potential adverse outcomes of ETS, including hypertension, respiratory morbidity, lung function, and asthma in children. **Conclusions:** The currently available evidence on prenatal ETS exposure in Chinese children has unveiled a wide range of health outcomes, including preterm birth, fetal development, behavioural disorders, and much more. However, Chinese studies in this area are still lacking and a gap still exists in relation to the strength of association between prenatal ETS exposure and some health risks. Efficient anti-smoking policies and smoking cessation programs should be developed to promote maternal and child health. Further research is also needed to provide better evidence in this field.

## 1. Background 

Environmental tobacco smoke (ETS), also known as second-hand smoke (SHS), contains more than 40 known or suspected carcinogens and various cardiovascular toxicants [1]. Early in the second half of the 1980s, evidence from several major international reports, including landmark publications issued by The International Agency for Research on Cancer, Australia’s National Health and Medical Research Council, the US Surgeon General, the US National Research Council, and the UK’s Scientific Committee on Tobacco and Health, showed that exposure to second-hand smoke increased the risk of illness and death in non-smokers from infancy to adulthood [2]. It is now well established that ETS exposure causes adverse health effects in thousands of passive smokers, including respiratory disease, cardiovascular disease, cancer, and mental and behavioural disorders [3,4,5].

Worldwide, approximately one-third of adult non-smokers and approximately 40% of children have been exposed to ETS at home, causing significant morbidity and mortality [6]. Pregnant women and children are particularly vulnerable to the harmful effects of second-hand smoke, as their bodies are undergoing developmental processes and they are more susceptible to the harmful substances in ETS and are at particular risk of serious health consequences [7,8]. Moreover, children are also generally unable to control their environment and have a lower ability to detoxify cancer-causing chemicals from smoke [7,8]. With numerous studies conducted to establish a strong association between ETS and adverse health outcomes in children, consistent and increasing evidence has shown that exposure to ETS during childhood and prenatally can contribute to adverse physical, psychological, and behavioural outcomes in children, such as respiratory tract infections (RTIs), asthma, low birth weight, orofacial clefts, childhood cancer, psychological symptoms, attention deficit/hyperactivity disorder (ADHD), and cognitive and language impairments [4,5,9,10,11,12].

China is the world’s largest consumer of tobacco, and exposure to ETS remains a significant public health problem [13]. Approximately 55.19% of non-smokers have been exposed to ETS in public places or at home [14]. Although policies prohibiting smoking in public places have been promoted globally over the past few decades with intensive research on the health effects of ETS, the household environment remains a high-risk setting for ETS exposure [13]. Non-smoking reproductive-age females are one of the high-risk groups, with 65.1% of Chinese women aged 15–49 years having been exposed to ETS in their homes according to the 2010 Global Adult Tobacco Survey (GATS) [15]. Moreover, subsequent and extended epidemiological studies have found that approximately 31.5% of children and 54.6% of pregnant women have been exposed to ETS at home in China [13,16]. It is well-established that in order to reduce the burden of disease attributable to second-hand smoke, an urgent need to reduce the prevalence of ETS exposure, especially among pregnant women and women of reproductive age, is warranted [6,8].

Despite all this, research on the association between children’s health outcomes and prenatal ETS exposure in China remains inadequate, and the lack of high-quality data from relevant studies hinders the development of more effective policies to prevent ETS exposure in children and pregnant women [13]. Therefore, a systematic scoping review was conducted to address this broad research question and represent the complex and heterogenous evidence [17]. This review aimed to identify, analyze, and summarize the literature on the adverse effects of fetal exposure to ETS on the health of children in China and identify gaps for further research.

## 2. Materials and Methods

A systematic scoping review was performed following the steps of the scoping review methodological framework developed by the Joanna Briggs Institute (JBI): (1) developing the research question; (2) identifying relevant studies; (3) selecting eligible studies; (4) extracting the results; and (5) presenting the results [18]. Following the Preferred Reporting Items for Systematic Reviews and Meta-Analyses Extension for Scoping Reviews (PRISMA-ScR) checklist [19], the reviewers presented the review results in a narrative format.

### 2.1. Eligibility Criteria

This study focused on children’s health outcomes with prenatal ETS exposure. Therefore, we included studies in which the population was fetuses or children ages 0–18 years whose mothers were exposed to ETS during pregnancy. ETS exposure was defined as indoor or outdoor exposure times daily more than 0 min/d. The studies included ETS exposure from various sites—at home, at the workplace, and in public places at any stage of the mother’s pregnancy. Additionally, since we were focusing on the context of China, only studies conducted in China were included. To avoid confusion of different terminology used and inaccuracy due to translation, we excluded studies published in languages other than English.

The specific inclusion criteria were as follow:Publication type: peer-reviewed journal article/reportArticle type: systematic reviews, meta-analyses, scoping reviews, cohort studies, cross-sectional studies, case-control studies, non-RCTs, case series, individual case reportsText availability: full textPublication date: since 2011Language: EnglishLocation: Mainland China, Hong Kong, Macau, and TaiwanStudy population: pregnant women, children ages 0–18 years, and fetus

### 2.2. Type of Resources

The literature search on the research topic was conducted using three electronic databases—PubMed, EBSCOhost, and ProQuest—and supplemented by other extensive search techniques, such as screening the reference lists of selected articles and checking the ‘cited by’ and ‘similar articles’ options in PubMed. This was done to include relevant articles not included in the primary search strategy.

To obtain the best and latest evidence and maximize the available data, we included all types of journal articles published since 2011. Firstly, experimental and quasi-experimental study designs were considered. However, randomized controlled trials were not be possible, as people cannot be randomly allocated to exposures. Moreover, epidemiological observational studies, such as prospective and retrospective cohort studies, case-control studies, and analytical cross-sectional studies, were the main types of sources that were included [20]. Furthermore, descriptive observational studies including case series, individual case reports, and systematic reviews that met the inclusion criteria were also considered [20]. However, this scoping review did not include qualitative studies. Finally, other reliable resources, such as government reports, were also considered for inclusion.

### 2.3. Search Strategy

A full search strategy was developed based on the initial limited search using the keywords contained in the titles and abstracts of relevant articles. Boolean operators were used to increase the sensitivity of the search. The keywords we applied included ‘environmental tobacco smoke’, ‘second-hand smoke’, ‘child*’, ‘foetal’, and ‘Chin*’. Moreover, search terms related to ‘prenatal exposure’ were added but then removed before the search process, as the search results were too limited when this keyword was applied. We refine the final search string as: (((environmental tobacco smoke) OR second-hand smoke) AND (((child*) OR foetal) OR fetal)) AND (Chin*). Then, in the filters area, “free full text”, “full text”, “10 years”, ‘English’, and ‘Humans’ were selected to further narrow the search results. The search process was carried out from June 2022 and revised again in June 2023 for the purpose of this submission.

### 2.4. Selection of Eligible Studies

The reviewers followed the PRISMA flow diagrams for study selection to identify the final included articles [19]. Firstly, as the search proceeded, articles retrieved from different databases that met the inclusion criteria were synchronized in Mendeley, then duplicate studies were removed. Then, the titles and abstracts were screened against the eligibility criteria by a reviewer (HE), while a second reviewer (XY) screened a random sample of 10% of the titles and abstracts. The results were in 100% agreement with the inclusion/exclusion decisions. Next, the study proceeded to a full-text review. Full texts of the selected articles were read through to exclude those that were not relevant to the research objective and those that did not meet the inclusion criteria. Finally, the reference lists of the retrieved articles were searched to include additional potentially relevant resources. Moreover, where no consensus was reached on inclusion/exclusion decisions, the two reviewers discussed it until an agreement was reached, or the senior author (FH) was asked to adjudicate.

An overview of the selection process is given in Figure 1.

### 2.5. Data Charting

The researchers divided the articles equally and extracted data from them, then re-examined the articles reviewed by the other, and finally summarized all the information. The extracted data were categorized and summarized in Excel and later exported into tables and graphs. The data extraction categories included the author(s), publication year, location, aims/purpose, methodology, study population, sample size, data collection and measurement, key findings that related to the research question, and research quality.

### 2.6. Critical Appraisal

All the included studies were assessed using the JBI Critical Appraisal tools for Systematic Reviews. Three different types of methodologies were identified in the selected articles, namely cohort, case-control, and cross-sectional studies. Consequently, the JBI Critical Appraisal Checklist for Cohort Studies, Checklist for Case-Control Studies, and Checklist for Cross-Sectional Studies were utilized to assess the methodological quality of various study methods. The process of quality appraisal was done by one reviewer (HY) and then reassessed by the second reviewer (XY) until a consensus was reached. No study was excluded in the quality assessment process. Each study received a score based on specific criteria in the checklists. The score was given as follows: (NO or unclear or not applicable) = 0 and (YES) = 1. Later, the total scores and the percentage of scores were calculated to evaluate the methodological quality. The maximum score was 11, and the low, moderate, and high quality scores fell within the ranges of 0–50%, 50–75%, and 76–100%, respectively.

### 2.7. Synthesis of Results

The study results were summarized and reported in narrative formats in two parts:A descriptive analysis, mapping the data and showing the distribution of studies by year of publication, origin, study method, aims, and quality.A thematic summary, narratively describing how the identified research relates to the systematic-scoping review research question and objectives, as well as the main findings from these organized by theme.

### 2.8. Characteristics and Quality of the Included Studies

The main characteristics of the selected articles are summarized in Table 1.

We included thirty journal articles, all of which were primary research. The majority (60%) were cross-sectional studies, five were cohort studies, and seven were case-control studies. The quality scores (%) of these studies, appraised using the JBI Critical Appraisal Checklists, ranged from 55% to 82%. The number of studies with high qualities, moderate qualities, and low qualities were 11, 19, and 0, respectively. The results of the critical appraisal are presented in Table 2.

### 2.9. Health Outcomes

Table 3 shows the main health outcomes of prenatal ETS exposure in Chinese children that were found in the twenty-two of the included articles.

### 2.10. Hypertension

One study by Zhang et al., (2020) [21] found a significant association between maternal ETS exposure during pregnancy and hypertension in their offspring. The research results showed that prenatal ETS exposure led to higher odds of hypertension in children, even after excluding the effects of potential confounders such as the age of the mother, prematurity, and low birth weight.

### 2.11. Foetal and Children’s Development

Seventeen studies reported the health impact of prenatal ETS exposure on fetal and child development.

### 2.12. Birth Outcomes

Eight articles examined the association between prenatal ETS exposure and children’s birth outcomes.

A cross-sectional study conducted in Shanghai investigated the association between LBW and mothers’ prenatal tobacco smoke exposure [22]. The results showed that the incidence of LBW was higher in children whose mothers were prenatally exposed to ETS, and their mean birth weight was 66.1 g lower than those without an exposure history. Furthermore, Huang et al. [33] found that prenatal ETS exposure was linked to a higher risk of giving birth to full-term low birth weight (FT-LBW) children, and the association remained consistent in subcategories of symmetric FT-LBW but not asymmetric FT-LBW. Moreover, another longitudinal prospective study suggested that maternal ETS exposure was likely to be an independent risk factor for fetal growth restriction and could reduce fetal birth length by more than 1 cm [48]. Additionally, exposure to ETS during pregnancy was associated with an increased risk of preterm birth (PTB) both before and after adjusting for potential confounders. Liu et al. [39] also found an increased risk of PTB among mothers with prenatal exposure to ETS, and the risk increased with the average level of daily ETS exposure. Surprisingly, the increased risk of PTB due to ETS during pregnancy was observed only among mothers who were more educated [28]. This might be because most participating mothers were well-educated, and there were significant differences in maternal educational level between PTBs and FTBs [39]. Liu et al. [32] also pointed out that paternal smoking during gestation had a significant association with PTB, and the association was more obvious in boys and children with old (≥34-year-old) mothers. Recent findings also suggested paternal smoking and preconception paternal smoking was independently positively associated with PTB risk, and the HRs increased with the increment of paternal smoking and preconception paternal smoking categories [46].

However, inconsistent findings were found in the association between ETS during pregnancy and PTB, LBW, and “small for gestational age” (SGA). One study reported a positive but non-significant effect of father smoking during pregnancy on infant birth outcomes [32], while another pointed out that the association between ETS during pregnancy and PTB was only found to be consistent in medically indicated PTB and late PTB, but not in spontaneous PTB and early PTB [28]. In addition, Chen et al. [48] stated that there was no evidence linking ETS to low birth weight or SGA based on their findings. Similarly, Lee et al. [47] found no significant association between infant birth weight and prenatal ETS exposure in their study. Chen et al. [28] also reported no significant association between ETS during pregnancy and the risk of LBW or SGA, and they suggested the inconsistent results could be due to different rates of LBW and SGA, differences in ETS definitions, and the exposure magnitude.

### 2.13. Orofacial Clefts (OFCs)

Two studies reported the health impact of prenatal ETS exposure on orofacial clefts. As Pi et al. [26] stated, prenatal ETS exposure among children of non-smoking mothers was significantly associated with an increased risk of developing OFCs. The association was dose-dependent; when mothers were exposed to ETS more than six days per week. There was an increased risk of OFCs in their offspring. Sakran et al. [44] also reported maternal passive smoking during early gestation to be a significant risk factor for nonsyndromic cleft lip and/or palate (NSCLP) incidence, with OR = 4.349.

### 2.14. Neural Tube Defects (NTDs)

One case-control study by Chen et al. [40] found a significant association between passive smoking and NTD occurrence, with a significant dose–response relationship between NTD risk and an exposure index of ETS or other household air pollution.

### 2.15. Congenital Heart Disease (CHD)

Three articles examined the association between prenatal ETS exposure and CHD, mostly focusing on exposure during the three months before pregnancy and in the first trimester of pregnancy. Both Song et al. [42] and Wang et al. [41] reported an increased risk of CHD in offspring whose mothers were exposed to SHS during the three months before pregnancy. All three identified articles showed significant associations between maternal exposure to ETS and CHDs in their offspring [42,45,46] also found a dose–response gradient between the risk of CHDs and maternal exposure to ETS in first trimester of pregnancy.

### 2.16. Developmental Coordination Disorder (DCD)

One case-control study conducted by Wu et al. [43] reported strong negative association between scores of the Chinese version of the Little Developmental Coordination Disorder Questionnaire (LDCDQ) and an increased risk of suspected DCD. Additionally, the prevalence of the suspected DCD was significantly higher in the prenatal SHS-exposed group, and the prevalence of suspected DCD in girls was higher than that in boys in the same age group.

### 2.17. Developmental Delay

Two studies explored the relationship between ETS exposure and developmental delay.

Ren et al. [27] noted that there was a higher rate of maternal exposure to ETS during pregnancy in children with cerebral palsy compared to the general population. After adjusting for confounding factors such as the delivery mode and birth weight, the results still revealed that children born to mothers exposed to ETS during pregnancy had a higher risk of cerebral palsy than the children of unexposed mothers, and this risk increased with increasing durations of ETS exposure [27]. Moreover, He et al. [4] study indicated that prenatal ETS exposure led to impairment in children’s early cognitive and language development in their first two years of life. Additionally, the level of language development was negatively correlated with the frequency of prenatal ETS exposure, with each additional pack of cigarettes smoked per week by a member of the household potentially associated with a 0.48-point decrease in early childhood language scores [4].

### 2.18. Behavioural Disorders

Six studies investigated the association between prenatal ETS exposure and children’s behavioural disorders.

### 2.19. ADHD

Three studies linked prenatal ETS exposure to the development of ADHD in children.

Wang et al. [12] examined prenatal tobacco smoking exposure (PSE) as a moderator in the association between genetic variants and ADHD. They reported that the genetic risk of ADHD may be affected by environmental factors, and the risk of genetic variation in ADHD is significantly increased if the child has been exposed to ETS prenatally. Thus, PSE was a potential risk factor for ADHD, which was significantly associated with all ADHD subtypes in children [12]. Lin, et al. [31] also found ETS exposure from pregnancy to childhood was associated with higher chances of having ADHD symptoms and subtypes, and the associations were stronger in the prenatal periods. Moreover, a significant positive association between prenatal ETS exposure and the risk of hyperactivity disorder in children was evident after adjusting for potential confounders in another cross-sectional study [24]. Women who have been exposed to ETS during any trimesters of pregnancy are more likely to have children who display hyperactive behaviours [24]. Furthermore, this relationship was dose-dependent; as the dose of ETS exposure increased, children were more likely to exhibit hyperactivity behaviours [24].

### 2.20. Autism Behaviour

One study showed that children with early life ETS exposure were more likely to exhibit autistic-like behaviours, and those who were exposed to ETS during gestation had a significantly increased risk of developing autistic-like behaviours [37]. This association persisted only in further analysis of the combined effect of children’s ETS exposure in three stages of early life—during pregnancy, from birth to one year, and from one to three years [37]. In addition, as the duration of exposure and the average number of cigarettes smoked in the child’s immediate environment increased, the risk of autistic-like behaviours also increased [37].

### 2.21. Other Disorders

Two articles pointed out the association between prenatal ETS exposure and behavioural disorders other than ADHD and autism. Liu et al. [23] investigated the association between prenatal ETS exposure and externalizing behaviours in Chinese children. Children born to mothers who had been exposed to ETS during pregnancy were at higher risk for externalizing behavioural problems, but no dose–response relationship was identified. In addition, a cohort study that estimated the associations between early ETS exposure during the prenatal and postnatal periods and several aspects of adolescent mental health found that prenatal ETS exposure from non-parental sources was associated with behavioural problems in children after adjusting for potential confounders [34]. While paternal smoking and maternal smoking were associated with more mental health problems, prenatal ETS exposure from non-parental sources, both occasional and daily, was associated with several behavioural problems but not with lower self-esteem or depressive symptoms [34].

### 2.22. Respiratory Outcomes

Four studies explored the association between prenatal ETS exposure and children’s respiratory outcomes.

In a study investigating the associations between children’s exposure to tobacco smoke in utero and in the first year of life and childhood and respiratory outcomes, Zhuge et al. [38] reported that 14.3% of children ages 3–8 years had at least one respiratory symptom. Pneumonia was the most frequently reported respiratory outcome, with a 32.3% lifetime incidence, followed by dry night cough (17.1%), frequent common colds (9.5%), and croup (6.0%). Except for frequent common colds, the crude odd ratios showed a stronger association between most respiratory health outcomes and paternal smoking only during pregnancy, and the effect of maternal smoking on respiratory outcomes was insignificant, possibly due to the low maternal smoking rate [38].

Similarly, Dong et al. [29] pointed out that compared to unexposed children, the prevalence of respiratory morbidities (including a history of asthma, current asthma, current wheeze, persistent cough, persistent phlegm, and allergic rhinitis) was higher in children with ETS exposure in utero, and increased with the increasing numbers of cigarettes smoked. Lee et al. [25] also suggested that perinatal and postnatal problems were more prevalent in the children whose mothers passively smoked during pregnancy than those whose mothers actively smoked during pregnancy and those that were unexposed. Fetal exposure to maternal passive smoking was significantly associated with having ever experienced wheezing, currently experiencing wheeze, and having ever experienced allergic rhinitis or eczema [25]. A dose–response relationship between having ever experienced wheezing and currently experiencing wheezing and increasing exposure to maternal passive smoking was also observed [25].

Furthermore, while the percentage of ETS exposure in utero among children with allergic predispositions was higher than that among children without allergic predispositions, children without allergic predispositions were more susceptible to ETS [29]. For children without allergic predispositions, ETS exposure in utero was associated with a history of asthma and current asthma only among boys [29].

Additionally, it was suggested that the adverse effects of maternal passive smoking or maternal active smoking on fetuses is due to their effects on lung function [25]. Hu et al. [30] found that in utero exposure to ETS was independently associated with decreased lung function. A significant association between ETS exposure in utero and decreased maximal mid-expiratory flow (MMEF) was found in children with asthma, but not in those without asthma, and in females but not in males [30]. No significant associations were observed between ETS exposure in utero, decreased forced vital capacity (FVC), and decreased absolute forced expiratory volume in 1 s (FEV1). However, it was reported that 66.4% and 57.2% of cases of prenatal ETS exposure with decreased FVC and decreased FEV1, respectively, were mediated by childhood asthma [30].

However, the association between children’s exposure to tobacco smoke in utero and respiratory outcomes were insignificant after adjusting for potential confounders [38]. The above study found that indoor smoke odor was clearly and strongly associated with most investigated respiratory outcomes. They argued that the perceived indoor smoke odor could be a more direct indicator of ETS exposure than parental smoking. This might be because parents who smoke may avoid smoking in the presence of children, which can be supported by the weak relationship between parental smoking and tobacco smoke odor [38].

## 3. Others

### 3.1. Astigmatism

One study investigating the association between ETS exposure during early life and early-onset astigmatism, showing no significant increased risk of astigmatism when children were exposed to ETS only during pregnancy. However, significant combined effects were observed—children were more likely to exhibit astigmatism when they were exposed to ETS during both pregnancy and from one to three years, or during both pregnancy and the first three years of life [35].

### 3.2. Sleep Disorders

According to Lin and his colleagues [36], ETS exposure during pregnancy was associated with higher total T-scores of the Sleep Disturbance Scale for Children (SDSC) and higher T-scores in six domains of sleep disturbance. ETS exposure during both pregnancy and the first two years of life had the highest total T-scores of SDSC and higher odds of increased sleep problems, with the strongest associations found in sleep–wake transition disorders (SWTD), as well as higher odds of long sleep latency in disorders of initiating and maintaining sleep (DIMS) [36].

## 4. Discussion

This systematic scoping review, which included thirty well-designed studies, has provided vital insights into the health impacts of ETS on Chinese children. Several studies have indicated that Chinese women have a low prevalence of passive smoking but a high incidence of ETS exposure at home and in public places. This special exposure profile reduces the interference of active smoking over passive smoking when investigating the health effects of tobacco exposure. Thus, our findings could provide convincing evidence for examining the harmful effects of ETS exposure during pregnancy on fetuses and children.

Prenatal ETS exposure has been reported to cause LBW, congenital birth defects, and infant mortality. According to Leonardi-Bee et al. [49], prenatal ETS exposure increases the risk of a fetus having congenital birth defects, such as cardiovascular, reproductive, musculoskeletal, and facial defects, by 10–50%. Maternal exposure to ETS during pregnancy was approximately twice as likely to result in LBW infants compared to non-exposed mothers [50]. Similar findings were shown in our study. Moreover, our review found that prenatal ETS exposure significantly increases Chinese children’s risks of developing respiratory diseases. Research has suggested that infant prenatal exposure to ETS increased the susceptibility to childhood respiratory diseases by impairing immune function in the early years of life [5]. As a result, ETS exposure affected lung development in infants and was associated with respiratory infections, wheezing, and asthma in children. In addition, it increased the risk of lifelong poor lung health and was associated with more serious respiratory diseases, such as lung cancer, in adulthood [5]. Our findings corroborated this conclusion. Furthermore, findings from this review also revealed that after adjusting for potential confounders, such as parental intelligence, parental literacy, and socioeconomic status, prenatal or childhood ETS exposure remained a significant contributor to impaired cognitive function and the onset of behavioural problems such as ADHD. These findings were consistent with similar findings from a systematic review by Zhou and colleagues [10].

By conducting this systematic review, we found numerous studies worldwide over the past decade investigating the health effects of prenatal ETS exposure on children. The currently available evidence on prenatal ETS exposure in Chinese children also has addressed a wide range of health outcomes, including respiratory diseases, hypertension, fetal and child development, behavioural disorders, sleep disorders, and astigmatism. However, relevant Chinese studies are still inadequate. For example, the research examining the association between ETS exposure, childhood cancers, and infant mortality in China remains almost nonexistent. Therefore, researchers need to focus on these gaps to design and conduct studies to further improve the well-being of children and pregnant women.

Evidence has shown a promising decline in the ETS exposure rate, from 46.8% to 30.8% in recent years, compared to data from the previous two decades [26]. However, the prevalence of ETS exposure in the household environment was still high [51]. Chen et al. [48] stated that although nearly half of women experienced passive smoking before pregnancy in a household environment, fewer women (36.5%) were exposed to ETS during their pregnancy from their family members. Meanwhile, ETS exposure before or during pregnancy is more common among women who are younger, less educated, multiparous, and have lower average personal income. Those women are also significantly more likely to drink alcohol, compounding their risk of poor pregnancy outcomes [28]. Therefore, all stakeholders must better understand and act effectively on ETS exposure in pregnant women.

The underlying biological mechanisms of how prenatal ETS leads to birth defects are still not fully understood. Tobacco smoke contains thousands of compounds, some of which are known to generally have toxic effects on reproductive health, such as carbon monoxide, nicotine, and metals. ETS contains similar constituents and may even have higher concentrations of some toxicants. A recent comprehensive Australian review has demonstrated how developing fetuses are particularly susceptible to the toxic compounds in tobacco [2]. Studies exploring the mechanisms of how smoking affects developmental outcomes mostly focus on the effects of those compounds, such as carbon monoxide, nicotine, and the carcinogens polycyclic aromatic hydrocarbon nitrosamines and aromatic amines.

To protect pregnant women from ETS exposure from their husbands or other household members, policymakers and public health professionals in the Chinese health system need to employ an evidence-based framework to assess and identify the health services needed for children and pregnant women and to develop more effective and tailored anti-smoking policies and smoking cessation programs for their family members. Moreover, healthcare providers, such as doctors and nurses, should educate both pregnant women and their husbands on the importance of tobacco control, raise public awareness about the harms of ETS exposure during pregnancy, and engage vulnerable populations in actively participating in smoking cessation programs.

## 5. Limitations

Our study had some limitations. Firstly, the number of included studies was small. Additionally, research articles published in Chinese were not included in this systematic scoping review, which may have led to data missing from potentially high-quality articles written in Chinese. Although we included only twenty-two articles, we carried out a well-structured screening process, and we included all the resources available to us under our screening criteria. This review involved studies conducted in urban and rural areas of Mainland of China, Hong Kong, and Taiwan. Thus, the findings can be broadly applied to the general Chinese population because the survey data were drawn from a diverse geographic and socioeconomic population in China.

Secondly, most studies measured the extent of ETS exposure by collecting self-reported data through questionnaires. However, people are often biased when they report their own experiences. For example, many individuals are either consciously or unconsciously influenced by “social desirability.” That is, they are more likely to report experiences that are socially acceptable or preferred rather than being truthful. In addition, the validity of self-reported data may be limited by the participants’ recall bias and introspective ability. Therefore, future studies on the association between ETS exposure and children’s health outcomes should use more objective exposure measurements to quantify exposure levels and reduce bias caused by data collection.

Finally, even though our systematic-scoping review study performed a quality appraisal for the included articles, which made the results of the review more reliable, further systematic reviews with meta-analyses are required to provide a better level of evidence and confirm some of the findings of this review.

## 6. Conclusions

This systematic scoping review provided a comprehensive summary of the adverse health effects of prenatal ETS exposure on Chinese children and identified a number of health issues in this context, including LBW, respiratory diseases, hypertension, fetal and child development, behavioural disorders, sleep disorders and astigmatism. Our findings may provide a strong framework for the development and implementation of smoking bans in indoor and public settings to minimize the harmful effects of ETS exposure on our offspring. In addition, the findings may also support health care providers in raising awareness about the above public health issues. A meta-analysis may be required to confirm these findings.

## Figures and Tables

**Figure 1 children-10-01354-f001:**
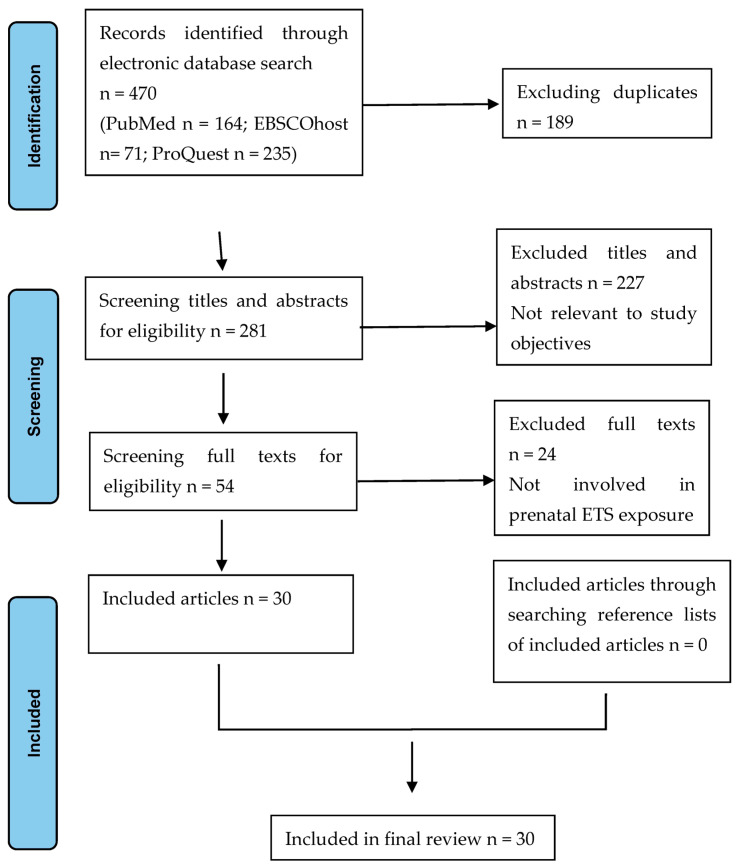
PRISMA flow chart of the study selection process.

**Table 1 children-10-01354-t001:** Characteristics and quality of the included studies.

Authors	Publication Year	Location	Aims/Purpose	Methodology	Sample Size	Study Population	Data Collection and Measurement	Study Quality
Zhang et al. [21]	2020	Liaoning Province	Evaluate the association of ETS exposure with hypertension and blood pressure (BP) in children.	Cross-sectional study	9354	School-aged children (5–17 years)	Questionnaire	Moderate
Chen et al. [13]	2020	Taiwan (Chaiyi)	Evaluated the influences of air quality, including ETS and particulate matter (PM), on fetal development.	Longitudinal correlation study	74	Non-smoking pregnant women	Questionnaire and laboratory result	High
Wang et al. [22]	2020	Shanghai	Explored whether the LBW in children is positively associated with mothers’ prenatal cigarette smoke exposure.	Cross-sectional study	8586	Kindergarten children	Interview and questionnaire	Moderate
Liu et al. [23]	2013	Jiangsu Province (Changzhou)	Examined the association between maternal ETS exposure during pregnancy and child behaviour problems.	Cross-sectional study	646	Mother-child pairs	Questionnaire and evaluation tool	Moderate
He et al. [4]	2018	Guizhou Province	Examined the association between prenatal exposure to ETS and the development of children in their first two years of life.	Cross-sectional study	446	Children	Questionnaire	Moderate
Lin et al. [24]	2017	Shenzhen City	Examined the association between prenatal ETS exposure and hyperactivity behaviours in young children.	Cross-sectional study	21,243	Preschool children	Questionnaire and evaluation tool	Moderate
Lee et al. [25]	2012	Beijing and Changchun Province	Examined the magnitude of association between maternal SHS exposure during pregnancy and reduction in infant birth weight in China.	Cross-sectional study	2770	Non-smoking postpartum women	Interview and questionnaire	Moderate
Wang et al. [12]	2019	Liuzhou City	Tested the hypothesis that prenatal tobacco smoking exposure (PSE) could modulate the association of genetic variants with ADHD.	Case-control study	401 (168 cases and 233 controls)	Children aged 6–12 years	Interview and questionnaire	High
Pi et al. [26]	2018	Shanxi Province	Examined whether exposure to SHS during the periconceptional period among nonsmoking women is associated with an increased risk for orofacial clefts (OFCs) in offspring.	Case-control study	1660 (240 cases and 1420 controls)	Newborns	Interview and questionnaire	Moderate
Ren et al. [27]	2020	Shandong Province	Assessed the association between maternal exposure to SHS during pregnancy and children’s cerebral palsy (CP).	Cross-sectional study	5067	Mother-child pairs	Questionnaire	Moderate
Chen et al. [28]	2021	Wuhan City	Clarified the association between ETS before and during pregnancy and the risk of adverse birth outcomes.	Cohort study	7147	Mothers-infant pairs	Questionnaire and delivery records	High
Dong et al. [29]	2011	Liaoning Province	Assessed the interaction between ETS exposure and allergic predisposition regarding respiratory health.	Cross-sectional study	23,474	Children from elementary schools	Questionnaire	Moderate
Hu et al. [30]	2017	Liaoning Province	Investigated whether gender or asthma status modifies the association between SHS exposure and lung function.	Cross-sectional study	6740	School-aged children	Questionnaire and electronic spirometers	Moderate
Lee at al. [25]	2012	Hong Kong	Examined the association between fetal exposure to maternal passive smoking and childhood asthma, allergic rhinitis, and eczema.	Cross-sectional study	7393	Children ≤ 14 Y	Questionnaire, interview, and evaluation tool	Moderate
Lin et al. [31]	2021	Liaoning Province	Evaluated the associations between prenatal, early postnatal, or current SHS exposure and ADHD symptoms and subtypes.	Cross-sectional study	45,562	School-aged children	Questionnaire and evaluation tool	Moderate
Liu et al. [32]	2017	Shanghai	Investigated the associations between household environmental factors during gestation and preterm birth, low birth weight, term low birth weight, and fetuses too small for their gestational age.	Cross-sectional study	13,335	Children ages 4–6 years	Questionnaire	Moderate
Huang et al. [33]	2020	Guangdong Province (Foshan and Shenzhen)	Examined the relationship between prenatal environmental tobacco smoke (ETS) exposure and full-term low birth weight (FT-LBW).	Case-control study	1632 (243 cases and 1389 controls)	Mothers–infant pairs	Interview and medical records	High
Leung et al. [34]	2015	Hong Kong	Estimated the associations between early SHS exposure during the prenatal and postnatal periods and several aspects of adolescent mental health.	Cohort study	7914	“Children of 1997” birth cohort	Questionnaire and evaluation tool	High
Li et al. [35]	2019	Shenzhen City (Longhua)	Investigated the association between ETS during early life and early-onset astigmatism.	Cross-sectional study	27,890	Preschool children	Questionnaire and medical diagnosis	Moderate
Lin et al. [36]	2021	Liaoning Province	Evaluated the associations between early-life SHS exposure and sleep problems in children.	Cross-sectional study	45,562	School-aged children	Questionnaire and evaluation tool	Moderate
Yang et al. [37]	2021	Shenzhen City (Longhua)	Explored the association between children’s exposure to ETS in early life and autistic-like behaviours.	Cross-sectional study	65,243	Preschool children	Questionnaire and evaluation tool	Moderate
Zhuge et al. [38]	2020	Urumqi, Taiyuan, Beijing, Nanjing, Shanghai, Wuhan, Chongqing, and Changsha	Analyzed the associations between ETS and dry night cough, croup, pneumonia, and the frequent common colds.	Cross-sectional study	41,176	Children ages 3–8 years	Questionnaires	Moderate
Liu et al. [39]	2022	Shenzhen City (Longhua)	Explored the independent and joint effects of prenatal exposure from multiple household air pollution (HAP) sources on PTB.	Cross-sectional study	63,038	Mother–child pairs	Questionnaire	Moderate
Chen et al. [40]	2023	Shanxi Province (Pingding, Xiyang, Shouyang, Taigu, and Zezhou).	Investigated the impact of maternal exposure to indoor air pollution from coal combustion and tobacco smoke on the risk for neural tube defects (NTDs).	Case-control study	739 (222 cases and 517 controls)	Women (during the periconceptional period)	Questionnaire and evaluation tool	High
Wang et al. [41]	2022	Rural China	Evaluated the association between paternal smoking and preterm birth (PTB).	Cohort study	5,298,043	Reproductive-aged couples	Questionnaire and medical diagnosis	High
Song et al. [42]	2022	Changsha, Hunan Province	Examined the role of the MTHFD1 gene and maternal smoking on infant CHD risk and investigated their interaction effects in Chinese populations.	Case-control study	968 (464 cases and 504 controls)	Mother–child pairs	Questionnaire and laboratory result	Moderate
Wu et al. [43]	2022	Mainland China (nationwide)	Investigated the association between prenatal SHS exposure and suspected DCD in preschoolers.	Cross-sectional study	149,005	Preschoolers	Questionnaire and evaluation tool	Moderate
Sakran et al. [44]	2022	Gansu Province	Identified the relationship between environmental factors and nonsyndromic cleft lip and/or palate (NSCL/P) in Northwest China.	Case-control study	1260 (600 cases and 660 controls)	Children and their parents	Interview	High
Deng et al. [45]	2022	West China Second University Hospital	Examined the association between maternal ETS and fetal CHDs and the potentially moderating effect of maternal hazardous and noxious substances (HNS), periconceptional folate intake, and paternal smoking.	Case-control study	1629 (749 cases and 880 controls)	Pregnant women	Questionnaire, interview, and medical diagnosis	High
Wang et al. [46]	2022	Central China	Estimated the associations between maternal active and passive smoking during the pre-pregnancy/early-pregnancy period and CHDs as well as its common phenotypes in offspring.	Cohort study	49,158	Pregnant women between the 8th and 14th weeks of gestation	Interview and medical diagnosis	High

**Table 2 children-10-01354-t002:** Quality scores of the included studies.

	Criterion											Total Score	Percentage (%)
Author (Year)	1	2	3	4	5	6	7	8	9	10	11		
**Cross-sectional studies**													
Zhang et al., (2020) [21]	1	1	0	1	1	0	1	1	n/a	n/a	n/a	6	55%
Wang et al., (2020) [22]	1	1	0	1	1	1	1	1	n/a	n/a	n/a	7	64%
Liu et al., (2013) [23]	1	1	0	1	1	1	1	1	n/a	n/a	n/a	7	64%
He et al., (2018) [4]	1	1	0	1	1	1	1	1	n/a	n/a	n/a	7	64%
Lin et al., (2017) [24]	1	1	0	1	1	1	1	1	n/a	n/a	n/a	7	64%
Lee et al., (2012) [47]	1	1	0	1	1	1	1	1	n/a	n/a	n/a	7	64%
Ren et al., (2020) [27]	1	1	0	1	1	0	1	1	n/a	n/a	n/a	6	55%
Dong et al., (2011) [29]	1	1	0	1	1	1	0	1	n/a	n/a	n/a	6	55%
Hu et al., (2017) [30]	1	1	0	1	1	1	1	1	n/a	n/a	n/a	7	64%
Lee et al., (2012) [25]	1	1	0	1	1	1	0	1	n/a	n/a	n/a	6	55%
Lin et al., (2021) [31]	1	1	0	1	1	1	1	1	n/a	n/a	n/a	7	64%
Liu et al., (2018) [32]	1	1	0	1	1	1	1	1	n/a	n/a	n/a	7	64%
Li et al., (2019) [35]	1	1	0	1	1	1	0	1	n/a	n/a	n/a	6	55%
Lin et al., (2021) [36]	1	1	0	1	1	1	1	1	n/a	n/a	n/a	7	64%
Yang et al., (2021) [37]	1	1	0	1	1	1	1	1	n/a	n/a	n/a	7	64%
Zhuge et al., (2020) [38]	1	1	0	1	1	1	0	1	n/a	n/a	n/a	6	55%
Liu et al., (2022) [39]	1	1	0	1	1	1	0	1	n/a	n/a	n/a	6	55%
Wu et al., (2022) [43]	1	1	0	1	1	1	1	1	n/a	n/a	n/a	7	64%
**Case-control studies**													
Wang et al., (2019) [12]	1	1	1	0	1	1	1	1	1	1	n/a	9	82%
Pi et al., (2018) [26]	1	0	1	0	1	1	1	1	1	1	n/a	8	73%
Huang et al., (2020) [33]	1	1	1	0	1	1	1	1	1	1	n/a	9	82%
Chen et al., (2023) [40]	1	1	1	0	1	1	1	1	1	1	n/a	9	82%
Song et al., (2022) [42]	1	1	1	0	1	1	1	1	1	1	n/a	9	82%
Sakran et al., (2022) [44]	1	1	1	0	1	1	1	1	1	1	n/a	9	82%
Deng et al., (2022) [45]	1	1	1	0	1	1	1	1	1	1	n/a	9	82%
**Cohort studies**													
Chen et al., (2020) [48]	1	1	1	1	1	1	1	0	1	n/a	1	9	82%
Chen et al., (2021) [28]	1	1	0	1	1	1	1	1	1	0	1	9	82%
Leung et al., (2015) [34]	1	1	0	1	1	1	0	1	1	1	1	9	82%
Wang et al., (2022) [41]	1	1	0	1	1	1	1	1	1	0	1	9	82%
Wang et al., (2022) [46]	1	1	0	1	1	1	1	1	1	n/a	1	9	82%

**Table 3 children-10-01354-t003:** Main health outcomes of prenatal ETS exposure in children.

Authors (Publication Year)	Main Results	Exposure Sources	Exposure Timepoint
1. Hypertension			
Zhang et al., (2020) [21]	Significant associations were observed between hypertension and ETS exposure in utero, with current major ETS exposure from fathers or anyone, and with intensity of ETS exposure greater than 1 cigarette per day. For SBP, significant associations were only observed in children with major ETS exposure from fathers and with cigarettes smoking >10/day.	Household (SHS from father or anyone)	In utero and within children’s first 2 years of life
2. Fetal and children’s development			
2.1. Birth outcomes			
Wang et al., (2020) [22]	The mean birthweight was 167.7 g and 66.1 g lower in children born to mothers with prenatal FHS and SHS exposure compared with those children whose mothers were not exposed, respectively.	Maternal passive and active smoking	Prenatal exposure
Huang et al., (2020) [33]	Significant association between prenatal ETS exposure and FT-LBW.	Maternal SHS exposure at home, workplaces, and public places	During pregnancy
Chen et al., (2020) [48]	ETS exposure decreased birth length by ≥1 cm, and is potentially an independent risk factor for fetal growth restriction.	Maternal SHS exposure at home, workplaces, and public places	During pregnancy
Chen et al., (2021) [28]	Significant association between exposure to ETS during pregnancy and PTB, but not LBW or SGA births.	Maternal ETS exposure (indoor and outdoor)	Before and/or during pregnancy.
Liu et al., (2018) [32]	Positive relation between paternal smoking during gestation and PB, LBW, and SGA, but not significant associations.	Household (paternal and maternal smoking)	During pregnancy
Lee et al., (2012) [47]	No deficit in mean birth weight was observed with exposure from all sources of SHS combined.	Maternal ETS exposure at home, workplaces, and public places	During pregnancy
Liu et al., (2022) [39]	Prenatal exposure to ETS increased the risk of PTB, and the PTB risk increased with the average level of daily ETS exposure	Household (Maternal exposure)	Prenatal exposure
Wang et al., (2022) [41]	Paternal smoking and preconception paternal smoking was independently positively associated with PTB risk. The HRs of PTB also increased with the increment of paternal smoking and preconception paternal smoking categories	Paternal and maternal smoking	Preconception exposure
2.2. Orofacial clefts			
Pi et al., (2018) [26]	Maternal SHS exposure during the periconceptional period increased the risk of OFCs in offspring among nonsmoking mothers, and there was a dose–response relationship.	Maternal SHS exposure at home and indoor public places	Prenatal exposure
Sakran et al., (2022) [44]	Maternal passive smoking was found to be a significant risk factor for NSCLP incidence.	Paternal and maternal smoking	The 1st trimester of gestation
2.3. Neural tube defects (NTDs)			
Chen et al., (2023) [40]	Significant association between passive smoking and neural tube defects (NTDs) including a dose–response gradient.	Household	During the periconceptional period (1 month before to 2 months after conception)
2.4. Congenital heart disease (CHD)			
Song et al., (2022) [42]	Increased risk of CHD in offspring whose mothers were exposed to secondhand smoke during the 3 months before pregnancy and in the first trimester of pregnancy	At home and/or at work/school	During the 3 months before pregnancy and in the first trimester of pregnancy
Deng et al., (2022) [45]	Maternal exposure to ETS in the first trimester was associated with increased risk of CHDs in a dose–response gradient.	Maternal exposure	In the first trimester of pregnancy
Wang et al., (2022) [46]	Significantly higher risks of CHDs were detected in offspring exposed to maternal cigarette smoking in the 3 months before pregnancy. Maternal cigarette smoking in early pregnancy was also independently associated with risk of CHDs in offspring.	Maternal exposure	During the 3 months before pregnancy and in early pregnancy
2.5. Developmental coordination disorder (DCD)			
Wu et al., (2022) [43]	Prenatal SHS exposure had a strong negative association with the total score of LDCDQ and increased the risk of suspected DCD	At home and/or at work	Prenatal and postnatal exposure
2.6. Developmental delay			
Ren et al., (2020) [27]	Children born to mothers exposed to SHS during pregnancy had a higher risk of CP, and the risk increased with exposure time.	Maternal SHS exposure	Prenatal exposure
He et al., (2018) [4]	Prenatal ETS exposure was associated with lower cognition scores and language scores, and the frequency of prenatal ETS exposure was negatively associated with language development before children reached two years old.	Maternal ETS exposure	Prenatal exposure
3. Behavioral disorders			
3.1. ADHD			
Wang et al., (2019) [12]	Prenatal tobacco smoke exposure was a significant risk factor for ADHD, even after adjusting for other potential confounders. The risk of the genetic variants in ADHD was increased significantly if the child had prenatal tobacco exposure.	Household and workplace	Prenatal and postnatal exposure
Lin et al., (2021) [31]	Significant association between SHS exposure from pregnancy to childhood and ADHD symptoms and subtypes.	SHS exposure	Prenatal, postnatal (i.e., first 2 years of life), and current periods
Lin et al., (2017) [24]	Prenatal ETS exposure was significantly associated with an increased risk of hyperactivity behaviours in young children, and there was a dose–response relationship.	Household (Maternal exposure)	Prenatal exposure
3.2. Autism Behaviour			
Yang et al., (2021) [37]	Significant association between being exposed to ETS during gestation and autistic-like behaviours, including a dose–response relationship.	Household	During pregnancy, from birth to one year, and from one to three years
3.3. Other disorders			
Liu et al., (2013) [23]	ETS exposure was associated with a higher risk of externalizing behaviour problems in the offspring of exposed mothers. However, it was not associated with internalizing or total behaviour problems. No dose-response relationship was found.	Maternal ETS exposure at home, the workplace, and other places	Prenatal exposure
Leung et al., (2015) [34]	Significant association between prenatal SHS exposure from non-parental sources and behavioural problems, and association between paternal smoking, maternal smoking, and mental health problems.	Non-parental and parental exposure	Prenatal and postnatal
4. Respiratory diseases			
Zhuge et al., (2020) [38]	Associations between most respiratory health outcomes and parental smoking, except for the frequent common colds. Stronger association for father smoking and insignificant effect for maternal smoking. Most association were insignificant after adjustment.	Parental smoking (mother only, father only, both)	During pregnancy, during the first year of life, and current periods
Dong et al., (2011) [29]	Significant association between ETS exposure in utero and the prevalence of respiratory morbidities, including a dose–response relationship.	Household (current and maternal)	Prenatal, postnatal (i.e., first 2 years of life), and current periods
Lee et al., (2012) [25]	Significant association between fetal exposure to ETS, having ever experienced wheezing, currently experiencing wheezing, and having ever experienced allergic rhinitis or eczema, including a dose–response relationship.	Household (maternal passive and active smoking)	During pregnancy
Hu et al., (2017) [30]	Significant association between in utero exposure to SHS and decreased lung function. Childhood asthma mediated the effects.	Household SHS exposure and maternal passive and active smoking	In utero and during early childhood
5. Others			
Li et al., (2019) [35]	No significant association between being exposed to ETS only during pregnancy and astigmatism. Significant associations were found between exposure to ETS during pregnancy and from one to three years, or during pregnancy, from birth to one year, and from one to three years old.	Household	During pregnancy, from birth to one year and from one to three years
Lin et al., (2021) [36]	Significant association between being exposed to ETS during pregnancy and sleep problems.	Household	Pregnancy and the first two years of life

## Data Availability

Data including studies being analysed in this review are available upon request.

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
