# Peer review of "The Impact of Prenatal Environmental Tobacco Smoking (ETS) and Exposure on Chinese Children: A Systematic Review"

_children, 2023, doi:10.3390/children10081354_

Round 1

Reviewer 1 Report

I suggest to the authors to consider the analysis of Chines smoker mothers in extra China Countries. In my wiew these  data are fondamental to compare the genetical and phenotipical aspect of the problem.

In the other countries the relation between smoke and intra uterin grow restriction is evident and Wellington demonstrated. Here is in dubt.  Why?

Finally in the other Countries is well note the association with sudden infant death syndrome. Here is not menzione. Why?

Author Response

Reviewer 1

Comment: I suggest to the authors to consider the analysis of Chines smoker mothers in extra China Countries. In my wiew these  data are fondamental to compare the genetical and phenotipical aspect of the problem.

Response: This is a good point raised by the reviewer and we thank the reviewer for raising it, however, we intended to only explore studies that focused on passive/ 2nd hand smoking in this systematic review. We gathered all evidence in relation to this focused topic and we believe the comparison would be valuable, however, may not necessarily be what we had in mind for this analysis and summary of evidence

Comment: In the other countries the relation between smoke and intra uterin grow restriction is evident and Wellington demonstrated. Here is in dubt.  Why?

Response: another great comment and we have somewhat addressed this in our review in the last paragraph of the Birth Outcome section. If more evidence is needed:

  • The evidence was sufficient to establish causal relationship between maternal active smoking during pregnancy and adverse effects on infant growth or an increased risk of LBW and SGA
  • The effect of prenatal ETS exposure is weaker compared to maternal active smoking, with a mean birth weight 60g lower for prenatal ETS exposure and average of 250g lower for maternal active smoking.
  • USDHHS (2004) also reported evidence to be “inadequate to infer the presence or absence of a causal relationship between maternal smoking and physical growth and neurocognitive development of children”, which was echoed in theexamination of second-hand smoke (USDHHS, 2006). 
  • Depend on the exposure magnitude, specific genotype of exposed women.
  • Other study in other country (Krishnamurthy et al., 2018) also showed no significant association between the SHS exposure of pregnant women and low birth weight.

Reference:

Krishnamurthy, A. V., Chinnakali, P., Dorairajan, G., Sundaram, S. P., Sarveswaran, G., Sivakumar, M., Krishnamoorthy, K., Dayalane, H., & Sinouvassan, V. (2018). Tobacco use, exposure to second-hand smoke among pregnant women and their association with birth weight: A retrospective cohort study. Journal of family medicine and primary care, 7(4), 728–733. https://doi.org/10.4103/jfmpc.jfmpc_269_17

US Department of Health and Human Services. (2004). The Health Consequences of Smoking: A Report of the Surgeon General. Atlanta: U.S. Department of Health and Human Services, Centers for Disease Control and Prevention, National Center for Chronic Disease Prevention and Health Promotion, Office on Smoking and Health; 2004.

US Department of Health and Human Services. (2006). The Health Consequences of Involuntary Exposure to Tobacco Smoke: A Report of the Surgeon General. Atlanta: U.S. Department of Health and Human Services, Centers for Disease Control and Prevention, National Center for Chronic Disease Prevention and Health Promotion, Office on Smoking and Health; 2006.

Comment: Finally in the other Countries is well note the association with sudden infant death syndrome. Here is not menzione. Why?

Response: please see above comment in relation to this question. Additionally, we tried to have the analysis limited to the research question in this systematic review and while we might have missed some aspects of health outcomes consistent with tobacco smoking, our summary of evidence here was dictated by the available research within the time frame of this review.

Reviewer 2 Report

Firstly, the inclusion criteria are very broad, encompassing a period from before birth to up to 18 years of age. However, there is a significant developmental difference between a human fetus and an 18-year-old adult. We kindly request that you provide a detailed explanation and justification for such a broad inclusion criterion in order to clarify its relevance to your study objectives.

Abstract, lines 19-20: The sentence appears to be incomplete and requires revision for clarity.

Please provide a clear justification for selecting 2011 as the starting point for inclusion criteria.

Methods, line 111: Consider correcting "foetal" to "fetus" for consistency.

Table 2: It would be more appropriate to include this table as an Appendix 1A for better organization and clarity.

Table 3: The table is lengthy and would benefit from being split into multiple tables. Consider presenting the data by the type of disorder to improve readability.

The Discussion section would benefit from including a subsection that explains the pathway between prenatal smoking exposure and the development of birth defects.

Dear authors, your manuscript would benefit from a revision by a native English speaker

Author Response

Reviewer 2

Comments and Suggestions for Authors

  • Firstly, the inclusion criteria are very broad, encompassing a period from before birth to up to 18 years of age. However, there is a significant developmental difference between a human fetus and an 18-year-old adult. We kindly request that you provide a detailed explanation and justification for such a broad inclusion criterion in order to clarify its relevance to your study objectives.

Response: This is a fair comment and we thank the reviewer for highlighting it. One of the reasons we applied this “broader criteria” was due to the following:

  • The underlying mechanism impact of ETS on health of the offspring is not fully understood. The exposure to tobacco smoke in utero could lead to many changes, even changes in DNA and potentially have impacts throughout the life course, including allergies, skin disease, impaired learning and memory, cognitive dysfunction, attention deficit hyperactivity disorder (ADHD), poorer academic achievement in childhood (and beyond), heart disease, type 2 diabetes, high blood pressure and being overweight in adulthood.

  • Also, 18 is the endpoint of puberty and it is the age at which a person legally become an adult in China. We would like to address the possible life-long effects on survival offspring, mapping the severity of threats on the vulnerable population group of children, to raise the public’s attention and strengthen their awareness of the importance of health promotion in tobacco smoking cessation.

  • Abstract, lines 19-20: The sentence appears to be incomplete and requires revision for clarity.

Response: we thank the reviewer for picking up this issue which we have now rectified- added words highlighted

  • Please provide a clear justification for selecting 2011 as the starting point for inclusion criteria.

Response: The review work commenced in late 2021. When we looked at the data they seemed to be scarce prior to the period of 10 years leading to 2011 so authors decided it made sense to include “last 10 years”. The work continued on the review beyond 2021 and so 2 more years were added making it 12 years all together. Additionally, it was not until January 2011 when China imposed nationwide policies of banning smoking from all confined public spaces such as restaurants, theatres, airplanes, trains, and buses.

  • Methods, line 111: Consider correcting "foetal" to "fetus" for consistency.

Response: Another great pick up by the author which is now corrected and highlighted, thank you (please also note that the word foetus is the Australian style while fetus is American)

  • Table 2: It would be more appropriate to include this table as an Appendix 1A for better organization and clarity.

Response: Thanks for this valuable comment- we thought about it and checked what other systematic reviews did with quality assessment tables and some had them in text and some as appendix. On this occasion we will leave this to the editorial team to decide. We are happy either way

  • Table 3: The table is lengthy and would benefit from being split into multiple tables. Consider presenting the data by the type of disorder to improve readability.

Response: Spot on, we also noticed that due to the level or precision in data and themes table 3 was a little hard to follow. Although we have seen complex tables in other systematic reviews, we have taken the reviewer’s comment on board and shaded the sections based on type of disorder which made it easier to follow. Let’s know if this helps and to avoid multiple tables in the one manuscript. Thank you

  • The Discussion section would benefit from including a subsection that explains the pathway between prenatal smoking exposure and the development of birth defects.

Response: This is anoher great point raised by the reviewer. We have no added a small section to highlight the link/ pathway between ETS and fetal development. See below (line 465-474 in paper):

The underlying biological mechanisms on how prenatal ETS leads to birth defects are still not fully revealed. Tobacco smoke contains thousands of compounds, some of which are known to generally have toxic effects on reproductive health, such as carbon monoxide, nicotine, and metals. ETS contains similar constituents and my even have higher concentration of some toxicants. A recent comprehensive Australian review has demonstrated how developing foetus are particularly susceptible to the toxic tobacco compounds (Greenhalgh et al., 2022). Studies exploring mechanisms on how smoking affect developmental outcomes mostly focus on the effects of those compounds, such as carbon monoxide, nicotine, and the carcinogens polycyclic aromatic hydrocarbons nitrosamines and aromatic amines.

  • Comments on the Quality of English Language- Dear authors, your manuscript would benefit from a revision by a native English speaker

Response: Thank you for the advice. This is now being done
